# Sensitivity Comparison of Refractive Index Transducer Optical Fiber Based on Surface Plasmon Resonance Using Ag, Cu, and Bimetallic Ag–Cu Layer

**DOI:** 10.3390/mi11010077

**Published:** 2020-01-10

**Authors:** Rozalina Zakaria, Nur Aina’a Mardhiah Zainuddin, Sofiah Athirah Raya, Siti Anis Khairani Alwi, Toni Anwar, Aliza Sarlan, Kawsar Ahmed, Iraj Sadegh Amiri

**Affiliations:** 1Photonics Research Centre, Faculty of Science, University of Malaya, Kuala Lumpur 50603, Malaysia; rozalina@um.edu.my (R.Z.); ainaamardhiah92@gmail.com (N.A.M.Z.); sofiahatirah@gmail.com (S.A.R.); sitikhairani96@gmail.com (S.A.K.A.); 2Department of Computer and Information Science (DCIS), Faculty of Science and Information Technology (FSIT), Universiti Teknologi Petronas (UTP), Seri Iskandar 32610, Malaysia; toni.anwar@utp.edu.my (T.A.); aliza_sarlan@utp.edu.my (A.S.); 3Department of Information and Communication Technology, Mawlana Bhashani Science and Technology University, Santosh, Tangail 1902, Bangladesh; k.ahmed.bd@ieee.org; 4Group of Bio-Photomatix, Mawlana Bhashani Science and Technology University, Santosh, Tangail 1902, Bangladesh; 5Computational Optics Research Group, Advanced Institute of Materials Science, Ton Duc Thang University, Ho Chi Minh City 700000, Vietnam; 6Faculty of Applied Sciences, Ton Duc Thang University, Ho Chi Minh City 700000, Vietnam

**Keywords:** D-shaped optical fiber, bimetallic layer, SPR, alcohol sensor

## Abstract

A single-mode optical fiber sensor uses surface plasmon resonance (SPR) with a bimetallic silver–copper (Ag–Cu) coating compared to a single layer of Ag and Cu itself. Bimetallic Ag–Cu sensors are constructed by simple fabrication on a side-polished optical fiber, followed by an electron beam evaporation of Ag and Cu films. For this investigation, the thickness of the single Ag layer was set to 30 nm and the single Cu layer was set to 30 nm; whereas for the bimetallic combined Ag–Cu layer the thickness of Ag was 7 nm and Cu 23 nm. The sensor performance was analyzed and compared experimentally and numerically using the COMSOL Multiphysics. A white light source was used with a broad optical bandwidth to provide a range of wavelengths to the optical fiber. The characteristics of the thin layers of Ag, Cu, and Ag–Cu as alcohol sensors were evaluated. We found that Cu was the most sensitive metallic layer compared to the Ag and the bimetallic Ag–Cu layers. For a 100% alcohol concentration, Cu showed a sensitivity of 425 nm/RIU followed by the bimetallic Ag–Cu layer with 108.33 nm/RIU, whereas the Ag layer was not detected. Interestingly, sensitivity reached saturation beyond the 20 nm thick layer of Ag. This shows that the Cu and the bimetallic Ag–Cu layers are suitable for an alcohol-based optical sensor.

## 1. Introduction

Gold and silver are the most widely used metals in a variety of surface plasmon resonance (SPR) optoelectronics devices [1]. The potential of SPR in surface characterization and monitoring of metal interfaces has been known since the late 1980s [2]. The first optical sensors were designed for the measurement of chemical and biological substances. Several types of optical fiber sensors have been investigated and used in various photonics communications [3,4]. They are designed to operate with interferometry and optical resonances. Such devices have used more advanced structures such as D-shaped optical fibers [5], long-period gratings [6], fiber Bragg gratings [7], as well as the photonic crystal fibers [8]. In addition, more advanced optical sensors with higher performance can be produced by integrating optical fibers and materials in the device. Surface plasmon resonance-based optical fiber technology with appropriate metal coating has also benefited from well-known innovations and flexibility in the device.

Surface plasmon resonance may be defined as the collective oscillation of free electrons on the metal-dielectric interface due to interaction with incident light. Resonance occurs when the frequencies of incoming photons and surface electrons match [9]. The incident angle, the input wavelength, the metal-dielectric functions, and the dielectric coefficient may affect the resonance conditions as can be demonstrated by the use of a prism-based SPR coupling. The use of optical fibers offers the advantages of light coupling without requiring angular adjustment of the incident light for SPR excitation. Because various metals have their own properties for specific practical sensing benefits, the use of Ag, Cu, and bimetallic Ag–Cu allow the resonance peak to be tunable, depending on the thickness of the layer and, thus, affecting the resonance of the sensor as well as the wavelength of each metal [10]. Single-mode fiber (SMF) refractive index mainly uses the fiber to passively guide light through a dielectric transducer that induces light amplitude modulation (transducer sensors).

Silver shows a sharp plasmon band in SPR and contributes greatly to sensing magnitude with better sensing resolution and is widely used in optical fiber sensors for a wide range of applications [11]. The electronic conductivity of Cu is significantly affected by the coupling effects of Ag. These can be used in various industrial applications that have been reported, for example, to produce an electrochemical sensor for the detection of 2-butanone and 1,2-dihydoxybenzene compounds [12].

The suitability of various metals used in optical fiber-based SPR has been theoretically reported by Navneet [13], who ranked the metal sensitivities by Au > Ag > Cu > Al. Copper shows the best signal-to-noise ratio (SNR) among these materials, with good_sensing performance at a thickness of 25 nm reported by Rana et al. [14]. In addition, the use of Cu as a chemical sensor offers an excellent choice as a functional biological probe [15]. Rotaru [16] has shown that Cu can be selectively metalized with double-stranded DNA on the oligonucleotide probe, which allows for the control of the cluster size and is very sensitive. As a matter of fact, the Ag-based sensor is known for its narrow spectral width but is also chemically unstable and is highly vulnerable to oxidation when used in liquid or gaseous environments.

We have experimentally demonstrated the use of both the capacity of different metals and the bimetallic combination layer to be used in an SPR optical fiber refractive index sensor at different concentrations of alcohol. The efficiency of the use of the active metallic layer covered by the oxide layer has also shown maximum sensitivity as reported by Singh et al. [17] using TiO_2_ followed by SiO_2_ and SnO_2_. The increased thickness also contributes to the increase in the size of the sensor as well as the additional layer to protect the metal from oxidation.

Due to the high sensitivity and high SNR, here, we observed a shift in the resonance wavelength by manipulating the refractive index of the sensing medium using Ag and Cu combined. In this work, we investigated a D-shaped SMF SPR sensor in which part of the jacket and cladding were polished in a D-shaped format and covered with a thin layer of Ag and Cu. We also assessed the best proposed thin metallic layer for the alcohol-based optical fiber sensor by keeping the polishing loss value as 1 dB and the thickness of the metal coating at 30 nm.

## 2. Numerical Simulation

We performed the computations using the finite element method (FEM) of the COMSOL Multiphysics simulation software, because it is simple to program due to the existence of a graphical environment with good accuracy. We applied the configuration of a D-shaped optical fiber which was used with a core and a cladding diameter of 8 μm and 125 μm, respectively. The theory for the perfectly matched circular boundary layer (PML) was applied to the structure in order to solve the Maxwell equations. The side-polished optical fiber, or D-shaped optical fiber, with following configurations; a thin layer of coated Ag with subsequent thin layer of Cu. Numerical analysis and calculations were carried out for the side-polished optical fiber structure as shown in Figure 1, where the arrow indicates the intensity of the distribution of the electric field near the fiber core.

In the resonance condition, the transmittance reaches its minimum value, while the field intensity reaches its maximum at the interface of the two layers, so that it is near the resonance angle where the surface plasmon wave generates a very strong evanescent field at the interface of the two layers. Figure 1b shows the dimensions of the D-shaped optical fiber captured from the microscope as the cladding remained at 3 μm with a loss of 1 dB.

It was also observed that the coupling point was enhanced by the thickness of Ag (30 nm) and Cu (10 nm) as shown in Figure 2. The pulse was initially propagated along the core region of the D-shaped fiber. Later, part of the optical energy traveled to the bimetallic region where absorption takes place, mainly due to the excitation of plasmon. As a result, the observation point moved from the core axis to the top layer of the fiber where the plasmon mode was applied which varied depending on the thickness of the coating and the interaction with the incident light.

Numerical analyses and calculations were carried out for the proposed side-polished optical fiber structure shown in Figure 1, where the indicator intensity of the distribution of the electric fields near the fiber core was omitted. The reason behind the Cu coating over the Ag layer is its proven ability to produce a high evanescent field while interacting with the propagation of light. It is also noted that the coupling point was enhanced by keeping the Ag thickness as 30 nm and the Cu thickness as 10 nm as shown in Figure 2. The thickness chosen here was 10 nm as the rounding off of the experimental data for better simulation observation. Changes in the transmitted power, which is a function of the refractive index as well as the resonance wavelength shift, determine the sensitivity of the device [2].

The COMSOL Multiphysics software used in this study is a powerful numerical simulation using FEM where FEM processes the structure of Maxwell’s vector equation as:(1)$$\Delta \times E\left(r,t\right)=-\frac{dB\;\left(r,t\right)}{dt}$$
(2)$$\Delta \times H\left(r,t\right)=-J\left(r,t\right)+\frac{dD\left(r,t\right)}{dt}$$
(3)Δ.D(r,t)=ρB(r,t)
(4)Δ.B(r,t)=0

Maxwell’s equations are evaluated by counting accurate measurements of adjacent subspaces, in which the finite number of air holes is divided into homogeneous subspaces for the selected cross-section. Full vector FEM-based numerical analysis is frequently used to simulate the electromagnetic area. The following FEM-based equation is evaluated from Maxwell’s equations using anisotropic PML [13,14].
(5)$$\nabla \times \left(\nabla \left[S^{-1}\right]\times E\right)-K_{o}^{2}n^{2}\left[S\right]E=0$$
where the wave number in the vacuum for the operating wavelength λ is denoted by $K_{0}=\frac{2\pi }{\lambda }$; *E* and *n* are the electric field vector and the refractive index of the medium, respectively. *S* and *S*^−1^, respectively, denote the matrix of PML and inverse matrix of PML.

Using the Cu coating over the Ag layer, where the bimetallic layer interacts with the propagated light, a highly evanescent field can be generated. With a 30 nm thick Ag layer and 10 nm thick Cu layer, the coupling effects can be improved theoretically as shown in Figure 2. The input pulse propagated within the bimetallic field, where plasmonic excitation induces absorption. As monitoring was carried out through the core to the top layer of the optical fiber, the plasmon may be affected by the incident light as well as by the thickness of the coating. The simulation was conducted for three cases (i.e., Ag, Cu, Ag–Cu) in conjunction with the experimental procedures. The proposed fiber was initially coated with Ag and the sensing substance applied over it. The distribution of the pulse initially started along the core region of the D-shaped fiber. Later, the portion of the optical energy travelled to the metallic region where absorption takes place, mainly due to the excitation of plasmon. As a result, the observation point turned from the core axis to the top layer of the fiber where the plasmon mode was exerted with its thickness coating and incident function. Sensing analyte, such as alcohol, was applied to the Cu layer where the light rays were transmitted in the sensing area and the interaction between the core and plasmon mode had occurred. Following this principle, the loss spectrum, $L_{c\;}\left[\frac{dB}{km}\right]=\frac{40\pi }{\ln\left(10\right)\;\lambda }Im\left(n_{eff}\right)\times 10^{3}$ for the different concentrations of alcohol, was calculated and plotted in Figure 2. Sensitivity was calculated by determining the resonance or peak wavelength of the total loss spectrum associated with the corresponding concentration of alcohol analytes. While the resonance points were indicated, the entire energy of the modes propagating in the core region of Ag, Cu or Ag–Cu could be trapped. The numerical value in Equation (9) was used to determine the sensitivity.

Figure 3a Shows the variation of the loss spectrum as a function of wavelength, the maximum coupling point at the metal–glass interface was inferred as the maximum loss, and the corresponding resonance wavelengths were calculated as 480 nm, 479 nm, 478 nm, and 476 nm for the alcohol concentrations of 0%, 20%, 60%, and 100%, respectively. The total sensing was reported as 164.0 nm/RIU. Similarly, the Ag layer replaced by the Cu layer was followed by the same principle as the calculated loss spectrum shown in Figure 3b,c. The corresponding resonance wavelengths for the alcohol concentrations of 0%, 20%, 60%, and 100% were obtained as 614 nm, 608 nm, 604 nm, and 602 nm, respectively. The total sensitivity was reported as 493.82 nm/RIU.

As shown in Figure 4, there was a phase-matching condition in which the maximum loss can be generalized with respect to the resonance wavelength. For alcohol concentrations of 0%, 20%, 40%, and 60%, the peak resonance wavelengths were 0.596 µm, 0.595 µm, 0.594 µm, and 0.593 µm, respectively. In other words, the resonance wavelength blue-shifted as the alcohol concentration increased.

Sensing was further optimized by increasing or decreasing the thickness of the Ag layer, whereas the Cu thickness remained constant. Figure 4 shows the spectral loss shift with respect to the Ag thickness, with the alcohol samples applied at 10% and 20% concentrations.

Varying the Ag thickness alters the spectral shift of the resonance wavelength with respect to the alcohol concentration. With a 10 nm Ag thickness, the resonance wavelength changed at 0.584 μm and 0.582 μm, respectively, for alcohol concentrations of 0% and 20%. With a Ag thickness of 20 nm, the resonance wavelength changes at 0.589 μm and 0.588 μm, respectively, with alcohol concentrations of 0% and 20%. The maximum difference of the resonance wavelength occurred when the Ag thickness was simulated at 10 nm. The interesting point is that beyond 20 nm, due to the constant thickness of the Cu layer, the effect of increasing the thickness of the Ag layer on the sensitivity was saturation. The strong evanescent was easily inferred from the loss spectrum figure for the bimetallic Ag–Cu combination. The loss absorption value was changed by adjusting the Ag thickness. In this instance, for the incremental thickness of the Ag layer, the peak values increased and the transmittance also changed. At this point, the greater the thickness of Ag, the confinement of the field and the coupling will be greater at the bimetallic–glass interface.

## 3. Methodology

A standard SMF consisting of a core and a cladding with a diameter of 9/125 μm was used. The D-shaped optical fiber was polished sideways. The laser source was connected to one end of the fiber and the optical power meter to the other end. We focused only on the effect of the remaining cladding (due to the consistent polishing loss of 1 dB), where the thickness of the metal coating was fixed at 30 nm. The source used for the input light had a transmission spectra between 400–1000 nm. It was inserted into one end of the fiber and the light spread to the other end. The experimental setup is shown in Figure 5.

It can be seen that the alcohol solution could be completely and quickly eluted, as the sensor’s response to the previous response was at 100% alcohol, giving the same dip in transmission at 60% alcohol concentration. A linear regression method was employed to analyze the relationship between sensor response and refractive index changes. This method calculated the best-fitting linear equation (straight line) for the data observed using the least squares approach. The slope of the linear fit to the measured sensor response at various surrounding refractive index changes was determined as the refractive index sensitivity of the fiber sensor under investigation.

This configuration of the side-polished optical fiber included Layer I which was a core fiber made of fused silica. According to the Sellmeier dispersion law, the variation of the refractive index of the fused silica, which is dependent on wavelength, can be described in the following equation:(6)$$n_{1}\left(\lambda \right)=\sqrt{1+\;\frac{A_{1}\lambda^{2}}{\lambda^{2}-B_{1}^{2}}+\frac{A_{2}\lambda^{2}}{\lambda^{2}-B_{2}^{2}}+\;\frac{A_{3}\lambda^{2}}{\lambda^{2}-B_{3}^{2}}}$$

The wavelength (λ) was expressed in µm and the Sellmeier coefficients were set to A_1_ = 0.6961663, A_2_ = 0.4079426, A_3_ = 0.8974794, B_1_ = 0.0684043 μm, B_2_ = 0.1162414 μm, and B_3_ = 9.896161 μm [2]. Here, the second layer (Layer II) was surrounded by a metallic layer. According to the Drude model, the dielectric constant of the metal can be described by:(7)$$\varepsilon_{m}\left(\lambda \right)=\varepsilon_{mr}+i\varepsilon_{mi}=1-\frac{\lambda^{2}\lambda_{c}}{\lambda_{p}^{2}\left(\lambda_{c}+i\lambda \right)}$$

Equation (7) shows two different wavelengths, *λ_p_* and *λ_c_*, representing the plasma and collision wavelength of the metal. The values of *λ_p_* and *λ_c_* for Ag were 1.4541 × 10^−7^ m and 1.7614 × 10^−5^ m, respectively. The values of *λ_p_* and *λ_c_* for Cu were 1.3617 × 10^−7^ m and 4.0852 × 10^−5^ m, respectively [3]. The sensing medium was referred to as Layer III. *ε_s_* represents the dielectric constant of the sensing medium and has a relation of *ε_s_* = *n_s_*^2^, where *n_s_* represents the refractive index of the sensing medium. For the excited surface plasmon wave, the resonance condition can be described by:(8)$$\frac{2\pi }{\lambda }n_{1}\sin\theta =Re\left\{k_{SP}\right\}$$
where $k_{sp}=\frac{\omega }{c}\sqrt{\frac{\varepsilon_{m}\varepsilon_{s}}{\varepsilon_{m}+\varepsilon_{s}}}$ = $\frac{2\pi }{\lambda }\sqrt{\frac{\varepsilon_{m}n_{s}^{2}}{\varepsilon_{m}+n_{s}^{2}}}$, and this represents the propagation constant of the surface plasmon, where the speed of light is *c*, and the incident frequency is *ω*. With an incident light angle of *θ*, the left part of Equation (8) interprets the incident light propagation constant, while the right side of the equation describes the actual part of the propagation constant of the surface plasmon [4]. Therefore, the difference in the refractive index can be measured by considering changes in the resonant wavelength.

## 4. Results and Discussion

Three important parameters indicate the performance of SPR sensors: sensitivity, SNR, and operating range, all of which need to be maximized. A shift in the refractive index will result in a shift in the resonance wavelength affecting the sensitivity of the SPR sensor. Sensitivity is as follows:(9)$$S_{n}=\frac{\delta \lambda_{res}}{\delta n_{s}}(\mathrm{nm}/\mathrm{RIU})$$

The SNR shows the accuracy of the sensor and can be determined from the width of the SPR curve. A narrow SPR, which means higher accuracy, may indicate a low dip in the SPR response. The SNR can be calculated by:(10)$$SNR\left(n\right)={\left[\frac{\delta \lambda_{res}}{\delta \lambda_{0.5}}\right]}_{n}$$
where $\delta \lambda_{res}$ is the shift in resonant wavelength and $\delta \lambda_{0.5}$ is the full-width at half maximum of the SPR curve. Various types of sensing mediums have been explored in optical fiber-based SPR sensors. For this work, the percentage of water diluted in alcohol was used on the basis of the refractive index tabulated in Table 1.

The resonance wavelength of each layer (i.e., Ag, Cu, and Ag–Cu) was observed for the media of different refractive indexes. The resonance wavelength was identified as a dip in the response curve of the sensor. The thickness of the metallic layer was set at 30 nm for all three cases. Transmission spectra as a function of wavelength were plotted for Ag, Cu, and bimetallic Ag–Cu thin layers with media of different refractive indexes. The transmittance spectrum of the SPR sensor versus wavelength of Ag, Cu, and Ag–Cu layers at a different refractive index of alcohol was recorded. Figure 6 shows that the shift in the transmittance dip of Ag at different alcohol concentrations initially remained at 484.68 nm (black line), shifted to 482.59 nm at 20% concentration (red line), shifted to 482.33 nm at 60% concentration (green line), and 100% concentration did not show any further shift (blue line). This indicates that Ag lost its sensitivity with a 100% alcohol concentration.

Figure 7 and Figure 8 show the dip in the transmission curve, which can be interpreted as the resonance wavelength shifts when the refractive index changes. The resonance wavelength underwent a blue shift as the refractive index of the alcohol transducer increased. As far as sensitivity is concerned, one has to see how much the resonance wavelength shifts for these metals. Metal resonance occurs primarily in the range of visible light within the wavelength range of 450–650 nm. The operating ranges are 450–550 nm, 600–650 nm, and 580–590 nm for Ag, Cu, and the bimetallic Ag–Cu combination, respectively. Total shift recorded (λ_diff_) for Ag was 2.35 nm (482.33–484.68 nm), Cu was 7.7 nm (609.64–617.34 nm), and bimetallic Ag–Cu was 0.78 nm (582.33–583.11 nm). Any measurement applied to its input will result in an output response, and this will provide a definition of the sensor response curve. Furthermore, the sensitivity curve can be determined by a derivative procedure of the response curve [18,19]. A unique feature of alcohol is that as its concentration in water increases by up to 80%, the refractive index also increases. If the alcohol concentration exceeds 80%, the refractive index drops. In the case of Cu and bimetallic Ag–Cu combination, the graphs of different alcohol concentrations follow a trend where, as the refractive index increased, the resonant wavelength shifted to the left. Sensitivity and SNR for Ag, Cu, and Ag–Cu SPR sensors are calculated as shown in Table 2, Table 3, Table 4 and Table 5. It is therefore worth noting that the bimetallic combinations are capable of adjusting the position as well as the width of the SPR curve depending upon which two metals are coupled. This is a useful design as an SPR sensor compared to a single metallic layer, because a single metal exhibits a trade-off between the two performance parameters and may provide the best sensitivity but the worst SNR. The bimetallic combination here gave a value higher than that reported by Sharma et al. (SNR = 0.071, Sn = 2.35 µm/RIU) [13] and our previous work using a single layer of Ag as alcohol sensor [20]. The effectiveness of the Ag–Cu combination proves the value of this configuration in various systems [21]. 

The response was the minimum intensity of the response curve and was the largest for the Ag sensor. Slope (sensitivity) and range values for the calibration curve for alcohol up to the concentration with the correlation coefficient obtained from Figure 8 are tabulated in Table 6. Slope uncertainty was the standard deviation and was greatest for the Cu sensor which could be due to the instability of the sensing process. A high linearity response of the regression line (correlation coefficient) indicates a good sensor. The linearity of the RI system can be plotted between the loss of resonance wavelength and the RI analyte. The observed value for the sensitivity of Cu and Ag–Cu in this work was higher than that reported by Subramaniam et al. [22], as it could be due to the sensing parameters, coating conditions, and sensing of alcohol itself.

The control of the surrounding refractive index with the use of alcohol solutions with various concentrations was investigated in this work. Various concentrations of alcohol solutions were used to measure the sensitivity of the D-shaped optical fiber sensor with a metallic overlayer. With the increase in alcohol concentration, the refractive index of the alcohol solution increased from 1.3450 to 1.3473, the transmitted light intensity of the fiber sensor exhibited a linear decrease in the resonance shift. The relationship between the resonance wavelength and the refractive index for an alcohol solution ranged from 1.3450 to 1.3473. Figure 9 shows the linear fit (correlation coefficient *R* = 0.7861 for Ag, *R* = 0.9145 for Cu, and *R* = 0.843 for Ag–Cu) to the sensor response plot as a function of the refractive index for the sensor fiber. The sensitivity of the Cu overlay showed the highest sensitivity corresponding to the calculation, as shown in Table 4, for Cu as 425 nm/RIU and 0.408 SNR, and the bimetallic combination had the lowest SNR ratio. The data do not show a satisfactory linear response here; however, this indicates that the combination of the metallic layer is most feasible for practical performance, especially in volatile samples, and shows how the sensitivity of single-layer sensors is easily lost when interacting with volatile samples. The satisfactory linear response *R*^2^ was close to one, and this shows that single-layer sensors have poor practical performance [23,24,25].

The results of this study clearly demonstrated that the interaction of alcohol with partially oxidized surfaces of Ag and Cu formed surface alkoxides and water after adsorption. Surface oxygen atoms increased the formation of alkoxide, and a dual site is suggested for the dissociative adsorption of alcohols on these surfaces.

Wang et al. [26] reported detailed studies of an XRD, TEM, and SEM methodology revealing that the new Ag–Cu bimetallic has, interestingly, an interaction between Ag and Cu that makes both metals more likely to lose electrons.

## 5. Conclusions

The working principle and the experimental results of the SPR-based D-shaped optical fiber refractive index sensor were presented. An SPR optical fiber refractive index transducer was designed and simulated using coatings of Ag, Cu, and Ag–Cu bimetallic combination. This study shows that each of these bimetallic combination coatings can produce an SPR peak and serve as an alcohol sensor better than our previous work in which a single metallic layer of Ag was used as an alcohol sensor. Copper showed the highest sensitivity, consistently tested with a 20% alcohol concentration. At higher alcohol concentrations, the resonant wavelength showed a blue shift, and a longer operating range can be achieved. The SPR side-polished fiber sensor presented here has several distinct advantages, such as a simple design and structure, low fabrication costs, and high biochemical performance. We conclude that the properties of the metal deposited on the optical fiber sensor can be controlled by selecting the metal as a single or bimetallic layer.

## Figures and Tables

**Figure 1 micromachines-11-00077-f001:**
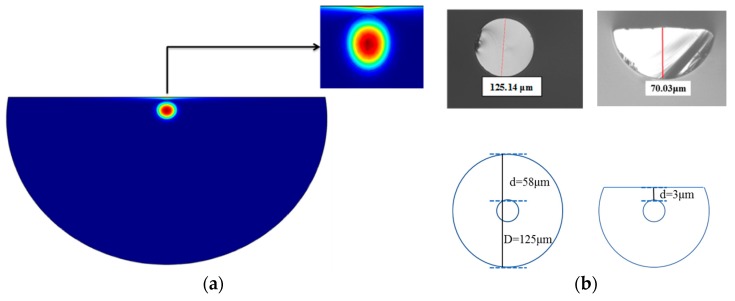
The cross-sectional view of the bimetallic D-shaped optical fiber at the coupling point of the metallic and core modes: (**a**) simulation model and (**b**) the dimensions of the D-shaped optical fiber from microscopy images.

**Figure 2 micromachines-11-00077-f002:**
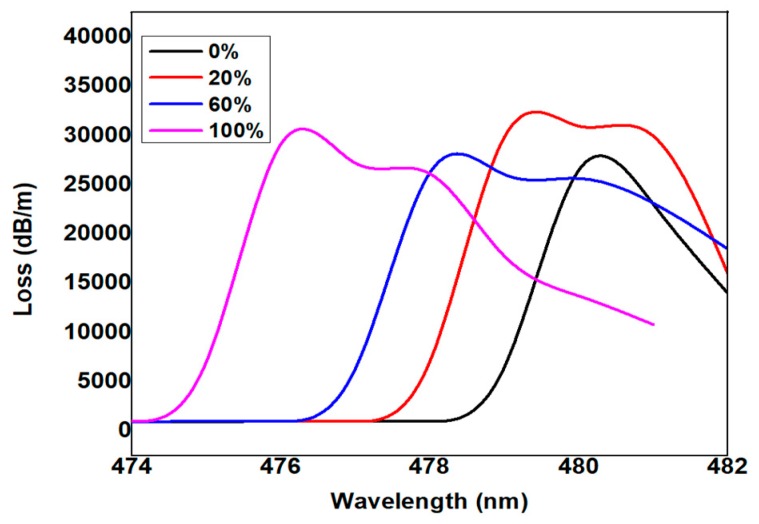
Loss spectra versus wavelength with respect to the variation of the alcohol concentration keeping Ag as 30 nm.

**Figure 3 micromachines-11-00077-f003:**
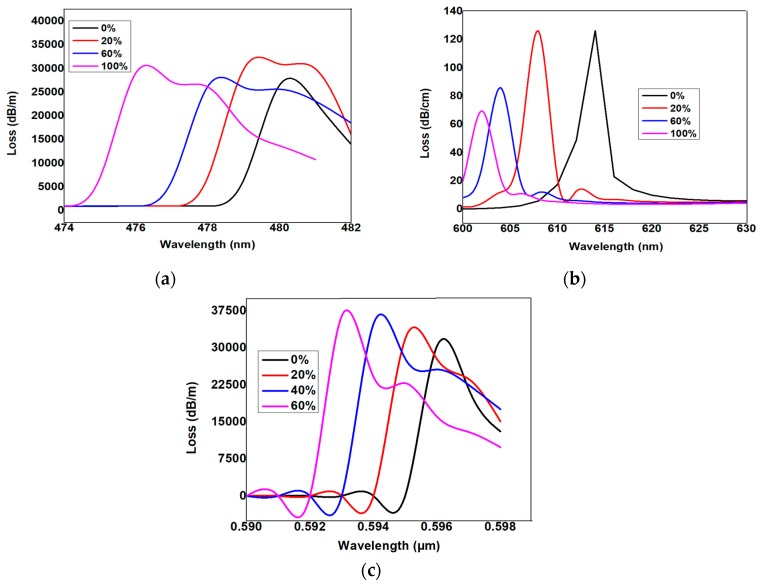
(**a**) Loss spectra versus wavelength with respect to the variation of the alcohol concentration keeping Ag as 30 nm. (**b**) Loss spectra versus wavelength with respect to the variation of the alcohol concentration keeping Cu as 10 nm. (**c**) Loss spectra versus wavelength with respect to the variation of the alcohol concentration keeping Ag and Cu thickness as 30 nm and 10 nm, respectively.

**Figure 4 micromachines-11-00077-f004:**
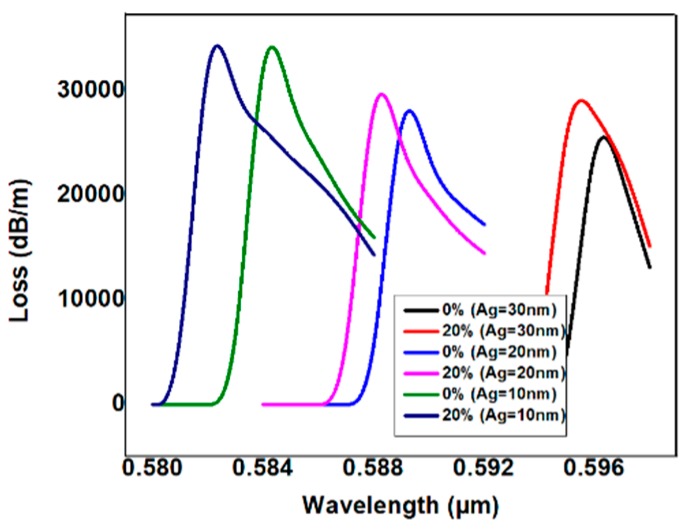
Variation of loss spectral shift for the bimetallic thickness of Ag = 30 nm, 20 nm, 10 nm under the constant thickness of Cu = 10 nm.

**Figure 5 micromachines-11-00077-f005:**
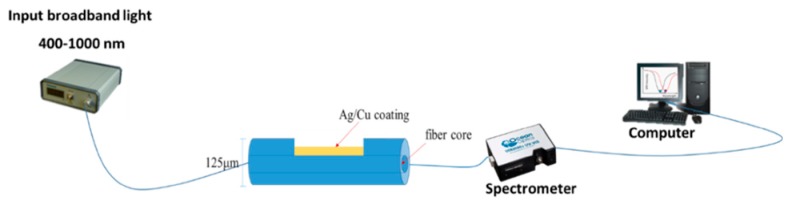
Schematic diagram of the D-shaped sensor covered with Ag, Cu and Ag/Cu.

**Figure 6 micromachines-11-00077-f006:**
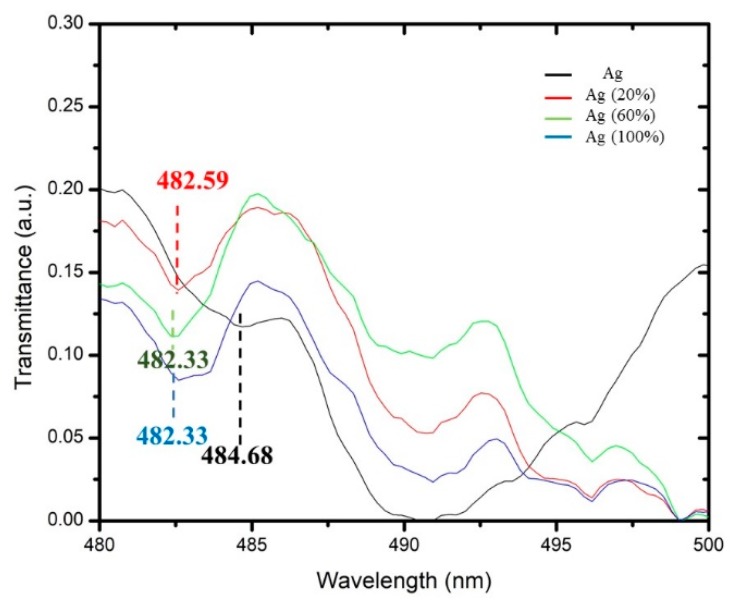
The surface plasmon resonance (SPR) sensor’s transmittance spectra versus the wavelength of Ag wavelength shifts at different alcohol refractive indexes, where (black line) Ag at a 484.68 nm (red line) wavelength of Ag 482.59 nm sensed at 20% of alcohol (green line); 482.33 nm of Ag sensed at 60% of alcohol; and (blue line) Ag sensed at 100% of alcohol concentration placed at the same wavelength as 60% of the alcohol at 482.33 nm.

**Figure 7 micromachines-11-00077-f007:**
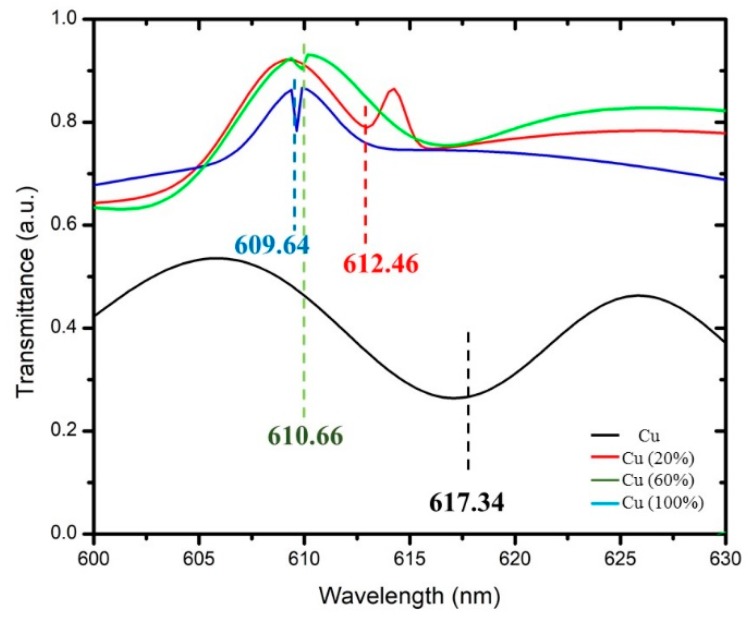
SPR sensor’s transmittance spectra versus the wavelength for Cu shifts at different alcohol refractive indexes as (black line) Cu at a 617.34 nm (red line) wavelength of Cu as 612.46 nm sensed at 20% of alcohol (green line); 610.66 nm of Cu sensed at 40% of alcohol; and (blue line) of Cu sensed using 60% of the alcohol concentration at 609.64 nm.

**Figure 8 micromachines-11-00077-f008:**
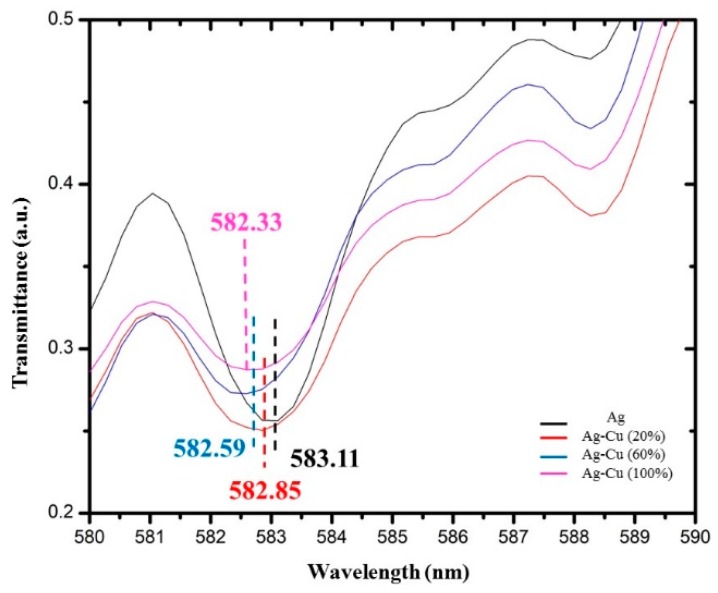
SPR sensor’s transmittance spectra versus the wavelength for Ag–Cu shifts at different alcohol refractive indexes as (black line) Ag–Cu at 583.11 nm (red line) wavelength of Ag–Cu as 582.85 nm sensed at 20% of alcohol (blue line); 582.59 nm of Cu sensed at 40% of alcohol; and Cu sensed using 60% of the alcohol concentration at 582.33 nm (pink line).

**Figure 9 micromachines-11-00077-f009:**
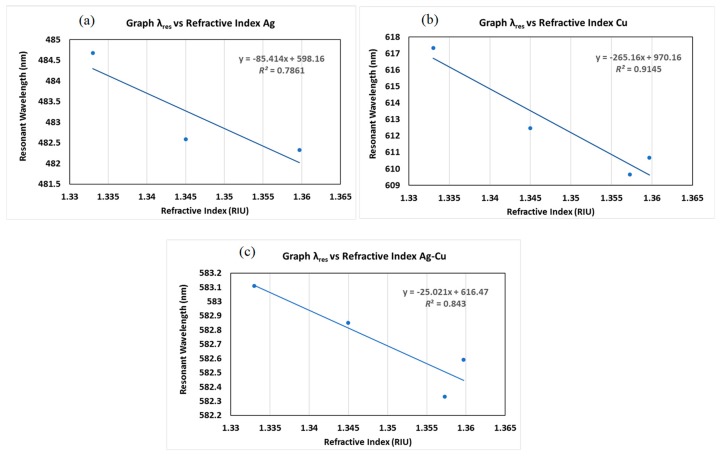
Graph of resonance wavelength against the refractive index for (**a**) Ag, (**b**) Cu, and (**c**) Ag–Cu bimetallic combination.

**Table 1 micromachines-11-00077-t001:** Refractive index of various concentrations of alcohol in water [17].

% of Alcohol in Water	RI at 30 °C
0	1.333
10	1.3384
20	1.3450
30	1.3510
40	1.3550
50	1.3578
60	1.3597
70	1.3608
80	1.3611
90	1.3603
100	1.3573

**Table 2 micromachines-11-00077-t002:** Ag–surface plasmon resonance (Ag–SPR) sensor: sensitivity and signal-to-noise ratio (SNR) comparison.

% of Alcohol in Water	Refractive Index (RI)	n_s_	λ_res_ (nm)	Δλ_res_ (nm)	Δλ_1/2_ (nm)	SNR	Sensitivity (nm/RIU)
0	1.3330	-	484.68	-	-	-	-
20	1.3450	0.012	482.59	2.09	1	2.09	174.167
60	1.3597	0.0147	482.33	0.26	1.5	0.173	17.687
100	1.3573	0.0024	482.33	-	-	-	-

**Table 3 micromachines-11-00077-t003:** Cu–SPR sensor: sensitivity and SNR comparison.

% of Alcohol in Water	RI	n_s_	λ_res_ (nm)	Δλ_res_ (nm)	Δλ_1/2_ (nm)	SNR	Sensitivity (nm/RIU)
0	1.3330	-	617.34	-	-	-	-
20	1.3450	0.012	612.46	4.88	2	2.44	406.667
60	1.3597	0.0147	610.66	1.8	1.5	1.2	122.44
100	1.3573	0.0024	609.64	1.02	2.5	0.408	425

**Table 4 micromachines-11-00077-t004:** Ag–Cu SPR sensor: sensitivity and SNR comparison.

% of Alcohol in Water	RI	n_s_	λ_res_ (nm)	Δλ_res_ (nm)	Δλ_1/2_ (nm)	SNR	Sensitivity (nm/RIU)
0	1.3330	-	583.11	-	-	-	-
20	1.3450	0.012	582.85	0.26	2	0.13	21.667
60	1.3597	0.0147	582.59	0.26	1.7	0.153	17.687
100	1.3573	0.0024	582.33	0.26	1.5	0.173	108.333

**Table 5 micromachines-11-00077-t005:** Comparison of the sensitivity, SNR, and SPR sensor for all metallic layers at 100%.

SPR Sensor	Signal-To-Noise Ratio (SNR)	Sensitivity (nm/RIU)
Ag	NA	NA
Cu	0.408	425
Ag–Cu	0.173	108.33

**Table 6 micromachines-11-00077-t006:** Properties of the metal-deposited SPR optical fiber sensors with Ag, Cu, and Ag–Cu at 100% of alcohol.

Metal	Response	Calibration Curve		
		Slope	Range (RI Units)	Correlation Coefficient
Ag	0.08 ± 0.04	−85.42	1.33–1.365	>0.7861
Cu	0.8 ± 0.12	−265.2	1.33–1.365	>0.9145
Ag–Cu	0.29 ± 0.09	−25.0	1.33–1.365	>0.843

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
