# Peer review of "Sensitivity Comparison of Refractive Index Transducer Optical Fiber Based on Surface Plasmon Resonance Using Ag, Cu, and Bimetallic Ag–Cu Layer"

_micromachines, 2020, doi:10.3390/mi11010077_

Round 1

Reviewer 1 Report

This paper presents a new sensing method with single mode optical fiber utilizing a bimetallic silver and copper coating. The performance of the sensor is simulated with numerical computation and demonstrated experimentally with different concentrations of alcohol solution. In overall, this research is an incremental technical work which has relatively low impact, but can be useful to the optical fiber sensing field if revised properly.

Major Revision:

The key technology of this research is the different metal cladding of the optical fiber. Can the author describe in more details of the fabrication of the optical fiber, especially the metal cladding layers? A flow chart with cartoons would be helpful.

Can the authors describe a little bit more in details why the Ag has less sensitivity comparing with Ag-Cu in theory?

For better understanding the manuscript, can the authors add a drawing model with the structure of the optical fiber in cross-section in Figure 1? In particular, the position of the metal layers need to be pointed in the model. Therefore, Figure 1 could probably has 2 sub-figures: (a) model, (b) simulation.

The organization and langrage of the manuscript is a little messy. Can the authors reorganize the manuscript with a clearer thought? The langrage also need to be improved.

Minor Revision:

Page 1 (Abstract):

Cu shows sensitivity of 425 nm …

=> 425 nm/RIU

Page 3 (2nd paragraph):

The reason behind the Cu coating over the silver layer is proven that it can able to …

=> either “can” or “is able to”

Page 5 (last 2nd paragraph):

… as alcohol concentrations rises.

=> the plural didn’t match

Page 7 (3rd paragraph):

Equation 2 …

=> Equation (7)?

Page 7 (5th paragraph):

… and they all need to be maximized A shift of …

=>   … and they all need to be maximized. A shift of …

Page 10 (1st paragraph):

Thus, it is a worth noted that …

=>    “a worth noted”  should be ”a worth noting”

Reviewer 2 Report

The Author provided a performance comparison analysis of surface plasmon resonance optical fiber refractive index sensors by coating a side polished D-shaped optical fiber with three different metallic coatings. The test solution for RI variation used is alcohol of different concentration while the coatings were single layer Ag, single layer Cu and bilayer Ag-Cu with all three coatings at 30nm thickness. It was reported by the authors that Ag-Cu bilayer coated sensor has a max sensitivity of 108.33/RIU.

General comments

Author should consider seeking help from native English speaker for proofreading as many parts of the manuscript was not clearly worded, causing difficulty in understanding the concept. Please check citation numbering as many of the citations in the latter half of the manuscripts are pointed to irrelevant work.

Suggestions

The claim that “Cu coating over the Ag layer … has been shown to produce stronger evanescent field” on line 95 needs to be supported either by simulation or citing references. It will be useful information for readers especially those who would like to replicate the result if the author describes how the D shaped fiber was fabricated with precision control of 1dB loss. Moreover, it will be critical to know the remaining cladding thickness. The use of alcohol as a test solution introduces error into the experiment. Despite the well-known RI of alcohol as tabulated in table 1, it is difficult to maintain the right concentration in real-world during the experiment due to the volatility of the substance itself. Hence, it is suggested to the authors that actual RI of the alcohol to be measured in the lab using appropriate equipment right before the experiment. Bimetallic layer thickness in the numerical analysis is inconsistent with actual implementation. Discussion on the rather large discrepancy of numerical analysis result and the experimental result should be included. Authors should specify the spectral analyzer used and its corresponding resolution. The observed spectral shift is small (sub nanometers), resolution of the equipment becomes important considerations. The linearity of the sensor, at least the bimetallic coating sensor should be evaluated with a test of smaller intervals. By visual inspection of figure 5 and 6, the single-layer sensors do not seem to exhibit a linear response. If so, the authors should elaborate. The claim on line 301 should not be based just on a 3-point analysis. If replication of study is not possible. it is advisable to remove claim on response linearity. Line 296 is misleading as a D shape optical fiber without coating can also function as a RI sensor operating with a different principle as a SPR sensor. Therefore, it is confusing to the reader to describe the proposed sensor as just a “D-shaped optical fiber sensor”.

Round 2

Reviewer 2 Report

Thank you for replying to my comments from the first review. Please consider including the discussions provided in your replies in the manuscript as much as possible to facilitate clear communication of your work. Please also seek assistance in improving the use of language in the writing. Best of luck. 

Author Response

Open Review

(x) I would not like to sign my review report

( ) I would like to sign my review report

English language and style

( ) Extensive editing of English language and style required

(x) Moderate English changes required

( ) English language and style are fine/minor spell check required

( ) I don't feel qualified to judge about the English language and style

Yes           Can be improved  Must be improved                 Not applicable

Does the introduction provide sufficient background and include all relevant references?

(x)             ( )              ( )              ( )

Is the research design appropriate?

(x)             ( )              ( )              ( )

Are the methods adequately described?

( )              (x)             ( )              ( )

Are the results clearly presented?

( )              (x)             ( )              ( )

Are the conclusions supported by the results?

(x)             ( )              ( )              ( )

Comments and Suggestions for Authors

Thank you for replying to my comments from the first review. Please consider including the discussions provided in your replies in the manuscript as much as possible to facilitate clear communication of your work. Please also seek assistance in improving the use of language in the writing. Best of luck.

Submission Date

15 September 2019

Date of this review

14 Oct 2019 16:03:20

Answer to reviewer:

Many thanks for your comments. All discussions provided earlier has been added in the manuscript mainly on page 11 with yellow highlights. I hope these comments will strengthen the arguments of comparing the use of single and bimetallic layer in this work. The used of single metallic layer of Ag as plasmonic material in detection alcohol is not an excellence choice as Ag interacts and loss sensitivity with volatile solutions. The used of Ag-Cu as an alternative layer is an appropriate option to overcome this issue.

The use of language has been checked by professional proofreading with certificate attached.

Open Review

(x) I would not like to sign my review report

( ) I would like to sign my review report

English language and style

( ) Extensive editing of English language and style required

(x) Moderate English changes required

( ) English language and style are fine/minor spell check required

( ) I don't feel qualified to judge about the English language and style

Yes           Can be improved  Must be improved                 Not applicable

Does the introduction provide sufficient background and include all relevant references?

(x)             ( )              ( )              ( )

Is the research design appropriate?

(x)             ( )              ( )              ( )

Are the methods adequately described?

( )              (x)             ( )              ( )

Are the results clearly presented?

( )              (x)             ( )              ( )

Are the conclusions supported by the results?

(x)             ( )              ( )              ( )

Comments and Suggestions for Authors

Thank you for replying to my comments from the first review. Please consider including the discussions provided in your replies in the manuscript as much as possible to facilitate clear communication of your work. Please also seek assistance in improving the use of language in the writing. Best of luck.

Submission Date

15 September 2019

Date of this review

14 Oct 2019 16:03:20

Answer to reviewer:

Many thanks for your comments. All discussions provided earlier has been added in the manuscript mainly on page 11 with yellow highlights. I hope these comments will strengthen the arguments of comparing the use of single and bimetallic layer in this work. The used of single metallic layer of Ag as plasmonic material in detection alcohol is not an excellence choice as Ag interacts and loss sensitivity with volatile solutions. The used of Ag-Cu as an alternative layer is an appropriate option to overcome this issue.

The use of language has been checked by professional proofreading with certificate attached.

This manuscript is a resubmission of an earlier submission. The following is a list of the peer review reports and author responses from that submission.

Round 1

Reviewer 1 Report

The paper reports on the realization of a fiber optic transducer based on Surface Plasmon Resonance phenomenon for detecting refractive index changes. The paper aims at optimizing the thickness of bi-metal layers for improving the performance of the device.   

The topic is appropriate for publication in the journal, however there are different points that weaken the technical and scientific soundness of the paper. In general, the paper is not so fluent or easy to be read, English need to be hardly improved,  figures are not so clearly explained. The final effect is getting the reader confused with respect to the real message of the paper.

Just to be more detailed:

-   English need a hard revision in the whole Introduction section and also in lines:78-84, 107-109, 120-122, 194-197, 214-216, 220-222, 233-234; 239-245.

-    As for the reported figures: it is not clear in what conditions the simulation reported in figure 1 is obtained.  Figure 2 is not well commented at all: the reader risks to be confused. Moreover it is not clear  why the proposed combination has been chosen. Figure 2 and 3 are  experimental or theoretical? Figures 5-6-7 need further attention: it is not clear to what transducer they are referred to; moreover figures a), b), c) are not visible.

-      As for alcohol solutions: they are aqueous solutions? RI values are measured or from literature? Some details here are missing.

-      Finally, important revisions from a scientific point of view are needed: authors tell about a sensor but actually they realized just a refractive index transducer,  because a sensing layer is not present, there is not reversibility or specificity tests and so on. So I would be less ambitious and presents things as they are. 

Moreover the authors confused the term sensitivity with Sensor response. Sensitivity can be obtained from the slope of a calibration curve (if linear in the investigated measurand range).

To this purpose I invite the authors to read some sensing parameter definitions in the following papers:

 A. D’Amico, C. Di Natale, A contribution on some basic definitions of sensors properties, IEEE Sens. J. 1 (2001) 183-190.

Q. Huang, D. Zeng, H. Li and C. Xie, Nanoscale, 2012, 4, 5651-5658.

In this sense, all the conclusions have to be revised accordinglgly.

For all the above comments I would suggest to accept the paper just after major revision.

Author Response

Please refer to my attachment. 

Reviewer 2 Report

In this work, the authors exploit the phenomenon of surface plasmon on metal dielectric interfaces for sensing the concentration in an alcoholic solution. However, the manuscript needs further discussion before being considered for publication. Results and discussion were not clearly and in-depth, and lacked the support of references. In my opinion, the authors fail to point out the novelty of their approach. This was because they were based on incorrectly interpreted data. The abstract, the discussion and conclusions are inconsistent with the results.

How did you determinate the sensor sensitivity? Include a graph with the obtained results.

Line 24 Delete “to”

Line 21 Change “of silver and goldfilm” for  “of silver and cooperfilm”

Line 28 sensitivity of 1741.67nm/RIU: from Table IV , it is clear that the real sensitivity of Ag is 174.167, and lower than the obtained for Cu (425).

Line 148-149 Authors mentioned: “The utilized light source as input light has the transmission spectra between 500-1400 nm and it is inserted into one of the fiber’s “. However, from Figure 1, the broadband light has a spectral range from 400 to 100 nm. Clarify.

In the figures 2 and 3, the scale between 37500 and 30000 dB/m is right? Authors should revise the Spectrometers Data Sheet. Generally, Spectrometers from Ocean Optics provide the redngs in “Counts”. It would be necessary to calibrate these equipments in order to obtain the proper dB or dBm.

In my opinion, this article is not appropriate for publishing in its current form.

Author Response

Please refer to my attachment. 

Round 2

Reviewer 1 Report

I am not completely satisfied by the revision of the paper that results still difficult to be read and with different inappropriate conclusions. It reveals a scarce knowledge of terminology and  analytical characterization of an optical transducer. To this regard, the term “sensor” used along the text is inappropriate as the analysis reveals that an optical transducer is proposed. Besides terminology, use of further sensing parameters along the text are reported inappropriately.

Different sentences and conclusions are not clear and does not find any confirmation in the reported figures.

More in detail:

Lines 91-92: The sentence: “It is was also observed that the coupling point was enhanced at when keeping the thickness of the Ag at 30 nm, and of the Cu layer at 10 nm, as shown in Figure 2.” is unclear from a scientific point of view and it does not find any link with the figure. Lines 98-109: the sentence is a tortuous and repeated reasoning that does not add meaningful information to the text. Lines 132-136: There is no evidence of a comparison with bare Ag to justify the beneficial role of Cu layer. Moreover the declaimed enhancement of coupling effects is not evident in the figure. The sentence “The input pulse propagates within the bimetallic region, where absorption occurs due to plasmonic excitation” is a redundant concept in the text that does not add any significant information for the threat of the reasoning. Lines145-450: comments of figure 2 are not clear from a scientific point of view. It is not clear how spectral loss are related to SPR resonances. Lines 158-164: the sentence show an evident inaccuracy in the data treatment and discussion. As explained in the previous revision, sensitivity parameter obtained with just 2 experimental point is an inappropropriate way to discuss experimental data. Lines 234-240: I do not find any correspondence between this comment and the reported figure. Spectra are too noisy to evaluate the declared spectral shifts. Moreover, it is not clear how the indicated resonance points have been identified; I would say that for black curve for example the resonance point is the minimum at 490nm. My comment can be applied both to figure 5 and 6.

It can be deduced that all the following comments related to those figures are missing their significance. I refer in particular to lines 262-271: information about total spectral shift is meaningless.

Comsol tool could be more useful here to model  theoretical transmittance spectra to be compared with experimental ones and identify the resonance spectral points unambiguously. The same for figure 7, where, moreover, the different investigated thickness are not explained at all.

Lines 280-290: sensitivity values calculated in tables II, III and IV are inaccurate. As explained you in my previous comments, calculating sensitivity with just two measurand points is inappropriate, especially if they are intended for comparison with analogues transducers. Table V in fact report “sensitivity” comparison in different refractive index range. Lines 297-302: The two sentences: When the concentration and, hence, the refractive index of an alcohol solution increased in the range of 1.3450–1.3473, the transmitted light intensity of the fiber sensor exhibited a linear decrease in resonance shift” and “It can be seen that the alcohol solution could be eluted completely and quickly, and the sensor response was back to the previous response as alcohol at 100% gives a dip of transmission as same as alcohol at concentration of 60%.”  do not find any experimental evidence in the text. Figure 8: it is not clear if the reported points are experimental or calculated; a linear regression with just three points and without any experimental error reported is a bit hazarded. Moreover, the sensitivity values that can be extracted here are far away from the ones reported in table II, III and IV.

Minor comments:

- Reference 13 refers to a different metal thickness with respect to the proposed thickness in the paper. So that it is not useful to be taken as a reference to build the discussion of the present paper.

- Reference 16 too is inappropriate with respect to the message of the paper.

-the term laser source is still confusing the reader.

-It is not clear if the percentage of alcohol solutions are in volume.

In general I have to say that the paper misses in satisfying  some important parameters needed for publication. For this reason  I suggest to reject the paper

Reviewer 2 Report

Every comment and suggestion made by the reviewer was taken into account and properly addressed. Thus, acceptance of this revised version is suggested.